# The reimbursement coverage decisions and pricing rules for medical devices in Taiwan

## Abstract

Medical technology is undergoing rapid transformations, and the classifications of medical devices have also expanded greatly; therefore, it is necessary to develop appropriate reimbursement policies and pricing mechanisms in a timely manner. This paper aims to introduce the reimbursement coverage and pricing rules for medical devices in Taiwan. In addition, this paper identifies and evaluates available health technology assessments (HTA) and literature on published websites concerning medical device decision-making processes and pricing systems in South Korea and Japan, which are near Taiwan and have similar reimbursement coverage processes. Reimbursement policy and pricing mechanisms are constantly being revised in Taiwan, Japan, and South Korea. Recently, all three countries attempted to establish new reimbursement coverage decision-making and pricing rules, adopting a differentiated approach based on the level of evidence required for the appropriated reimbursement in terms of a feasible evaluation mechanism for providing patients with more effective medical devices. This article is expected to contribute to providing references to new reimbursement coverage decision-making and pricing rules.

Hsin-Yi Tsai[1]
Yu-Wen Huang[2]
Shu-Ya Chang[2]
Li-Ying Huang[1]
Chii Jeng Lin[3,4]
Po-Chang Lee[2,5]

1 Division of Health Technology Assessment, Center for Drug Evaluation, Taipei, Taiwan

2 National Health Insurance Administration, Ministry of Health and Welfare, Taipei, Taiwan

3 College of Medicine, National Cheng Kung University, Tainan, Taiwan

4 Medical Devices Reimbursement Committee of Taiwan, Tainan, Taiwan

5 Departement of Surgery, College of Medicine, National Cheng Kung University, Tainain, Taiwan

## Introduction

Medical technology is undergoing rapid transformations, and new medical devices are constantly being introduced. In Taiwan, the number of medical device items listed by its National Health Insurance (NHI) have also increased, surpassing 10,000 devices in 2020 [1]. The classifications (e.g., materials, functions, specifications) have also increased.

Medical devices are being developed more rapidly than pharmaceuticals, yet evidence of real clinical efficacy is difficult to obtain in a short time; for example, the true efficacy of a bioresorbable vascular scaffold cannot be demonstrated until 3 years after implantation [2]. Related benefits are also difficult to reflect in clinical evidence (e.g., safer for user, improvements in treatment procedures), and the classification is more complicated than pharmaceuticals. Therefore, it is inappropriate for medical devices and pharmaceuticals to be governed by the same rules.

Generally speaking, every country has a diverse healthcare system and unique challenges related to patients' access to medical devices. In Australia and the United State, there are two types of health insurance systems: "public and private health insurance" [3], [4]. However, Canada and three Asian countries, Taiwan, South Korea, and Japan, have universal healthcare systems that cover almost all medical expenses. These three Asian countries have similar reimbursement mechanisms for medical devices whose funding can be classified as technical fees, separate from technical fees, and unreimbursed. South Korea and Japan serve as reference countries for Taiwan. Although Taiwan, Japan, and South Korea have similar reimbursement mechanisms for medical devices, they still have somewhat different payment systems. Both Taiwan and South Korea have two main insurance payment systems for medical devices:

1. a fee-for-service schedule and
2. a diagnosis-related group (DRG) system [5].

Meanwhile, Japan is paid on a fee-for-service basis. Before reimbursement approval decisions can be made in Taiwan and South Korea, some medical devices need to be reviewed using health technology assessments (HTA); this requirement does not apply to Japan. Table 1 provides an overview of the healthcare systems in these three Asia countries.

In light of the limited information about the reimbursement coverage and pricing rules regulating medical devices from Taiwan, the nation recently attempted to adopt a new policy in pricing rules for medical devices. Therefore, this paper will describe the process for determining the reimbursement policy and pricing mechanisms for medical devices in Taiwan. In addition, medical device decision-making processes and pricing systems in South Korea and Japan, which have similar reimbursement coverage processes as Taiwan, will also be examined.

# Reimbursement coverage decisions and pricing in detail

## Taiwan

Taiwan established its universal NHI program in 1995 and implemented the second-generation NHI in 2013 [6]. When manufacturers apply the new function medical device for national health insurance reimbursement, the National Health Insurance Administration (NHIA) requests that the Center for Drug Evaluation/Health Technology Assessment (CDE/HTA) assesses the clinical effectiveness and provides an economic evaluation, such as budget impact analysis for some designated cases (including drugs, medical services, and medical devices; medical devices were started in 2011) [7]. For example, if a medical device's budget impact analysis shows an increase of more than 30 million New Taiwan dollars (NTD) in NHI, new functional categories need to be reviewed by CDE/HTA [8]. This review process also includes national HTA reports from the United Kingdom, Canada, and Australia as references to assist in making decisions for NHI reimbursement listings at the Pharmaceutical Benefit and Reimbursement Scheme (PBRS) Joint Committee meeting, with final approval being granted by the Ministry of Health and Welfare (MOHW) [9], [10].

Several items are not covered under the NHI program, as per Article 51 of the NHI Act, including dentures, artificial eyes, spectacles, hearing aids, wheelchairs, and canes [11]. Two main payment systems for medical devices exist:

1. a fee-for-service schedule and
2. Taiwan diagnosis-related groups (Tw-DRGs) [5].

In addition, some special medical devices, called special devices, tend to be far more expensive than existing similar items in the NHI's fee schedule, so Taiwan implemented its balance-billing system to ease the NHI's financial burden and provide patients with more choices. If a medical device falls into the fee-for-service or Tw-DRGs,

NHI fully covers the cost of the medical device; however, if the medical devices fall into balanced-billing, NHI covers only part of the price, and the consumer pays the remaining amount [12]. Balance-billing items must have evidence supporting that they meet at least one of the following criteria:

1. more durable,
2. more convenient for patients,
3. easier to monitor,
4. more compatible with specific equipment or instruments, or
5. custom-made to be more comfortable.

As of this writing, Taiwan distinguishes 9 categories of balance-billing items, including pacemakers with additional functions, drug eluting coronary artery stent, special materials of hip prosthesis, special function artificial intraocular lenses, special materials of bio-prosthetic heart valve, programmable ventriculoperitoneal shunt, drug device combinational products for superficial femoral artery stenosis, ablation catheter for treatment of complicated cardiac arrhythmia, and special materials of extended gamma nail [13].

Two kinds of application categories exist for special devices:

1. existing functional category and
2. new functional category (including innovative and improved functional categories).

If the medical devices refer to a label that has been approved by the Taiwan Food and Drug Administration (TFDA) and is the same as an item already listed in the PBRS, they should be classified in the existing functional category (e.g., implantable cardioverter defibrillator); however, if a new medical device shows improved clinical function compared to the best medical device with the basic or similar function listed in the PBRS, it can be classified as an improved functional category (e.g., implantable defibrillator with a conditional intended use in an MRI environment) [14]. If clinically supportive evidence submitted by manufacturers demonstrates that the new special device is a breakthrough or is innovative, manufacturers can request the creation of an innovative functional category (e.g., transcatheter leadless pacemaker system) [15]. The manufacturer also needs to provide a budget impact analysis when it requests coverage under a new functional category. A more detailed process for reimbursement listing applications is provided in Figure 1 [16].

Three classifications for medical device reimbursement schedules exist:

1. funding under a technical fee, which is called general material (such as disposable consumables during treatment, including sutures);
2. separate from the technical fee, which is for special devices (implantable/specific non-implantable) that can be fully reimbursed (e.g., an implantable cardioverter defibrillator) or balance-billed; and

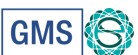

Table 1: Overview of the healthcare systems in Taiwan, Japan and South Korea [5]

| | Taiwan [10], [34] | Japan [10], [21], [22] | South Korea [31] |
|---|---|---|---|
| Regulatory system | TFDA | PMDA / MHLW | MFDS |
| Healthcare system | NHI: single payer | NHI: multiple payer | NHI: single payer |
| HTA | CDE/HTA[a] | HTA | NECA[e] |
| Medical device reimbursement approval process (in order) | 7. NHIA<br>8. HTA<br>9. Expert meeting<br>10. Patient participation<br>11. PBRS meeting<br>12. MOHW | 4. MHLW<br>5. Medical Material Expert Group meeting<br>6. Chuikyo<br>7. MHLW | 5. NECA<br>6. HIRA<br>7. MDEEC<br>8. MOHW |
| Payment system of medical device | DRG[b]<br>Fee for service[c] (balance-billing system)[d] | Fee for service | DRG<br>Fee for service |

TFDA, Taiwan Food and Drug Administration; NHI, National Health Insurance; CDE, Center for Drug Evaluation; HTA, health technology assessments; NHIA, National Health Insurance Administration; PBRS, Pharmaceutical Benefit and Reimbursement Scheme; MOHW, Ministry of Health and Welfare; Chuikyo, Central Social Insurance Medical Council; DRG, diagnosis-related group; PMDA, Pharmaceuticals and Medical Devices Agency; MHLW, Ministry of Health, Labour, and Welfare; MFDS, Ministry of Food and Drug Safety; NECA, National Evidence-based Healthcare Collaborating Agency; HIRA, Health Insurance Review and Assessment; MDEEC, Medical Device Expert Evaluation Committee.

[a] Budget impact analysis increases more than 30 million NTD in NHI, new functional categories, or others.
[b] DRG payment system is a bundled system covered with the payment fees, which consist of all goods and services provided during hospitalization.
[c] Fee for service is a payment system where services are unbundled and paid for separately, such as coronary stent system and titanium spinal, among others.
[d] To be eligible for balance-billing items, they must (i) be more durable, (ii) be more convenient for medical procedures (iii) show better clinical efficacy.
[e] Only new devices associated with a new technique or without comparable products must be reviewed by the nHTA.

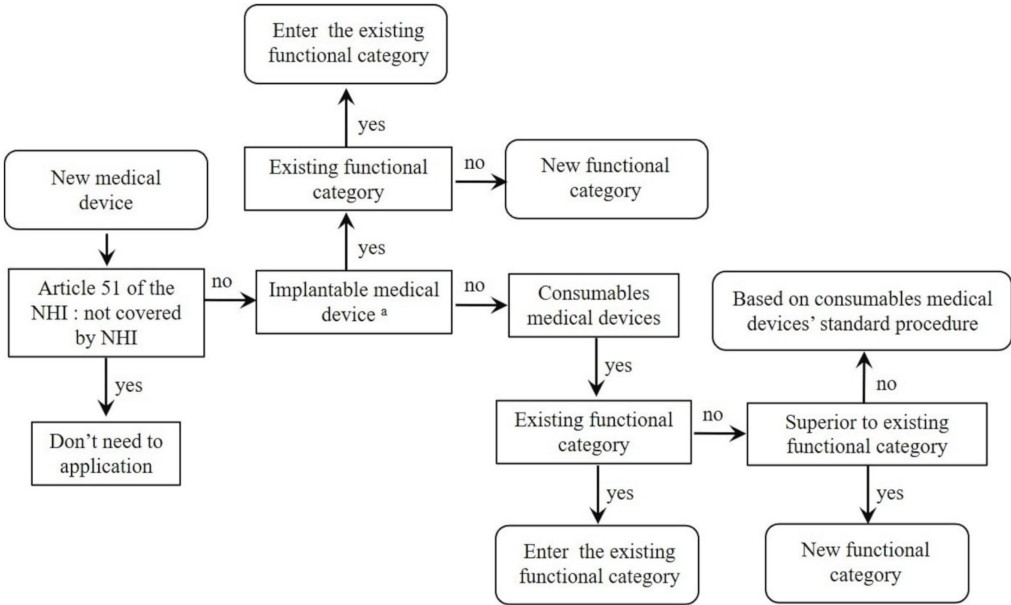

[a] Non-implantable devices superior to existing consumable devices (such as extracorporeal membrane oxygenation, drug-coated balloon) or that remain in the body for a short time (such as removable skin staple) could be listed as special devices.

Figure 1: Process for reimbursement-listing applications [16].

3. not (or not yet) reimbursed items (patients pay out of pocket; e.g., ligament) [5], [10], [17].

However, some implantable devices are not yet listed for reimbursement and should be assigned self-paid codes by the NHIA. The reimbursement point for a new functional category is discussed in the PBRS Joint Meeting; if the special device falls under an existing functional category, the reimbursement point should be reported to the PBRS

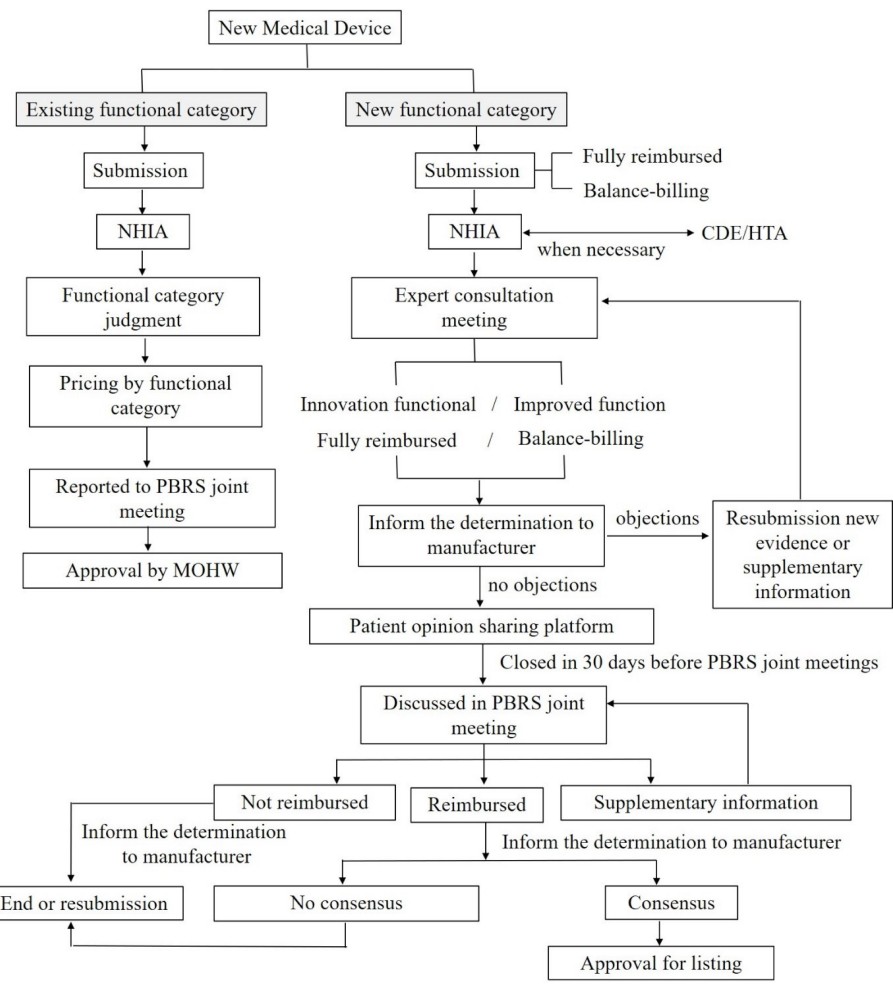

NHIA, National Health Insurance Administration; CDE, Center for Drug Evaluation; HTA, health technology assessments; PBRS, Pharmaceutical Benefit and Reimbursement Scheme; MOHW, Ministry of Health and Welfare

**Figure 2: Flow chart of reimbursement determination for new medical devices [8], [10], [16]**

Joint Meeting [8]. Figure 2 presents a flow chart of reimbursement determination for new medical devices. The reimbursement price for medical devices is adjusted using a price–volume survey (PVS), usually over a 4-year cycle or every 2 years for balance-billing and new functional category items [10], [18].

Some challenges to medical device reimbursement in Taiwan still exist, such as scarce resources with healthcare costs, political pressures, new technologies, and innovative medical devices. Furthermore, some manufacturers prefer to keep special devices in the self-paid market to generate higher profits, and they do not want to import special devices for people with rare diseases or newborns. The NHIA has attempted to amend the reimbursement schedules, such as those paid under a Tw-DRG-based scheme [10]. In addition to these challenges, the NHIA has faced other critical issues, including

1. self-pay medical devices items and categories becoming more complicated;
2. the price of self-pay medical devices lacking transparency for the same functional medical devices, with significant differences in the pricing in each hospital; and

3. insufficient information provided to patients to enable them to choose appropriate special devices.

Recently, the NHIA attempted to reduce the self-pay items and adopt reasonable pricing rules. In 2020, the NHIA amended the balance-billing ratio rule for special devices and adopted a differentiated level of clinical evidence for the appropriate balance-billing ratio and patients' payment price. Special devices superior to existing items based on evidence from randomized controlled trials, meta-analyses, or cohort studies may gain up to 40% payment — the highest balance-billing ratio from NHI; however, for cases with only control studies, international conferences with peer reviews, or case series (=10 cases), the balance-billing ratio from NHI is less than 20%. In other words, items supported by more clinically beneficial evidence can receive more health insurance benefits from NHI, resulting in patients paying less, while items with insufficient documentation for clinical effectiveness will remain in self-pay markets. In addition to the clinical evidence submitted by the manufacturer, high-value balance-billing items are evaluated through HTA to ensure their clinical effectiveness and safety while

providing patients with cost-effectiveness for special medical device items.

Furthermore, the NHIA amended the criteria for the balance-billing items, requiring them to have evidence supporting the claim that they

1. are more durable,
2. are more convenient for medical procedures, or
3. have superior clinical effectiveness.

A transcatheter aortic valve implantation (TAVI) served as a pilot study for this balance-billing program [19]. However, because of unmet medical needs, the TAVI was ultimately listed as a fully reimbursed item with strict criteria in 2020. The NHIA set up the new revision website to provide a comparative price for self-pay medical devices to help patients choose an appropriate medical device. The information on the website includes (1) the item and charge fee of fully self-pay or balance-billing items; (2) replaceable devices from NHI items; and (3) the reason for not reimbursing payments [20].

## Japan

Japan's NHI system was implemented in 1961 [21]. Reimbursement usually takes 6 to 9 months once a product is approved, although depending on the reimbursement category it can sometimes take more than one year [22]. The Japanese government announced the formal start of the HTA scheme (Cost Effectiveness Evaluation) in April 2019 for price adjustments (downward or upward) but not for reimbursement decisions [23], [24]. Japan introduced the Diagnosis Procedure Combination (DPC) system, which is similar to the DRG, but surgery-related costs (including medical devices) are still paid on a fee-for-service-based system [2], [21].

Japan has two types of reimbursement rules for medical devices. Special Designated Treatment Materials (STMs) are separate from the technical fee and include most of the high-cost, medium- to high-risk single-use devices that are disposable or implantable (e.g., pacemakers and artificial discs). Meanwhile, non-STMs include low-risk medical devices (classified by the US Food and Drug Administration as Class I and II), which could be low-priced, reusable, and durable diagnostic devices included in the technical fee but not eligible for an individual reimbursement price (e.g., magnetic resonance imaging or intraocular lenses) [2], [5], [21], [25].

Reimbursement prices for new STMs are determined by the Ministry of Health, Labour, and Welfare (MHLW) according to the functional category system. In this system, functional category criteria are based on their structure, purpose of use, efficacy, and effectiveness/performance [5]. Therefore, STMs in the same category have a specific reimbursement price [21]. The manufacturer can submit the STM to the MHLW to categorize the new medical device into an existing functional category covered by reimbursement; however, if the new medical device and/or its corresponding technical fee does not meet the existing functional category, the manufacturer must apply

to the MHLW to create a new functional category and new technical fee to acquire reimbursement for the device [5], [21]. Different reimbursement assessment categories exist:

1. A1, A2, and A3 categories are covered within the technical fee;
2. B1, B2, and B3 categories of STMs are not bundled into the technical fee; and
3. C1 and C2 categories are similar to the B category, where C1 is the establishment of a new functional category of STMs without technical fees and C2 is the application for new technical fees or a new functional category and new technical fees at the same time [21], [26].

If the medical device is not suitable for NHI reimbursement, it will be classified as F; devices in this classification may be extremely innovative and the technology involved is still not fully developed or the risk is higher than the clinical benefit [27]. Japan also has a unique system called Advanced Medicine (Senshin Iryou) that offers highly technological medical care. Such services are not covered by public health insurance [28]. Recently, the Japanese government indicated that, if a new medical device can replace an old device, new and old devices can be used in the same population, and there is no need to set the new functional category for a replaceable device. For example, the pacemaker and implantable cardioverter-defibrillator (ICD) categories for use in MRI scans can be merged with the pacemaker and ICD categories that cannot be used in MRI scans [21].

Two types of price calculation methods for new STMs exist:

1. the similar function category comparison method and
2. the cost accounting method [25].

Medical device prices are revised every 2 years based on market prices, foreign average pricing (the United States, the United Kingdom, Australia, France, and Germany), profitability, and category restructuring [2], [21], [22], [29].

Japan faces some issues for medical device reimbursement. For example, if clinical efficacy data are not ready, a company cannot get innovation premiums even by providing evidence of clinical efficacy after reimbursement. Japan established a new Challenge Application in 2018 to allow manufacturers to submit new clinical data (efficacy or safety; only for STMs) to request the creation of a new category or premium price after the reimbursement is listed [2], [21]. This new policy attempts to address the lack of evidence of medical devices at the time of the reimbursement listing and develop the most feasible pricing rule.

## South Korea

South Korea also has an NHI system [5]. After approval from the Ministry of Food and Drug Safety (MFDS), any new device associated with a new technique (i.e., never

introduced before) and without comparable products must be reviewed by the new HTA (nHTA) at the National Evidence-based Healthcare Collaborating Agency (NECA) before applying for reimbursement. Most devices do not meet the criteria for requiring review by nHTA. Manufacturers must submit reimbursement applications to the MOHW or Health Insurance Review and Assessment Services (HIRA) within 30 days of MFDS approval [30], [31], and reimbursement approval usually takes about 6 months [22]. Similar to Taiwan and Japan, South Korea has three different reimbursement mechanisms for new medical devices:

1. separate from the technical fee,
2. bundled into the technical fee (usually consumables, such as syringes), and
3. not reimbursed [31].

For new medical devices for which reimbursement is separate from the technical fee, the reimbursement price is usually decided after comparing the new device with an already listed product in the same functional category including similar items and purposes. The functional category (including indication for use and three physical characteristics: material, shape, and size) usually decides how an item is classified. For example, regardless of the manufacturer and brand, all pacemakers are classified into the same functional category [10], [31]. Fee-for-service is the main payment method in South Korea. In general, payment is 70% from the government and 30% from the patient; however, in cases of medical devices without clear clinical benefits or a lack of cost-effectiveness, patients need to pay more, ranging from 50% to 80% [32]. The DRG payment system (DRG costs include medical services, drugs, and consumables) was introduced in January 2002, but it has only been applied to seven procedure groups [30].

In 2015, South Korea adopted a new reimbursement coverage and pricing rule using multi-criteria decision analysis (MCDA) to require appropriate premium reimbursement pricing; manufacturers can submit clinical evidence or technological and functional information to determine appropriate premium reimbursement pricing [33]. New medical device reimbursement prices may be increased by an additional 10% to 100% based on their clinical efficacy, cost-effectiveness, and technological innovation compared to similar products [10]. If relevant supporting clinical evidence exists and the clinical efficacy can be statistically significantly improved, the reimbursement is determined by value appraisal standard (I), which can result in up to a 100% premium price. However, if the supporting evidence is obtained from a technical document submitted to the MFDS, the reimbursement price will be determined through value appraisal standards (II) adding only up to 50% of the premium price [10], [32], [33]. If the newly introduced medical device is in a new category with no corresponding appropriate comparable product, the reimbursement price will consider the existing treatment cost of similar diseases (cost-accounting calculation method) or the price in other countries, manufacturing costs (import costs), merchandising price, and other factors [10], [32]. In addition, reimbursement price adjustment includes two different reimbursement mechanisms:

1. the actual transaction price system and
2. the foreign exchange rate mechanism, which may be implemented every 6 months, in April and October [31].

However, South Korea does not have clearly defined criteria for single-use devices (SUDs) funded under the technical fee category. In addition, there is a lack of transparency and consistency in the decision-making process. Without a proper review of the clinical benefits from new SUDs or the management of the costs of outdated treatment procedures, new SUDs are easily and inaccurately classified into an existing category. In addition, the cost of new SUDs is significantly higher than their corresponding surgical fees, which is not properly reflected in the procedure fee, leading to inappropriate reuse problems. Despite approval by the MFDS, most innovative medical devices are classified as unreimbursed because of the lack of evidence related to cost-effectiveness [5], [31].

## Summary of medical device reimbursement and pricing mechanisms in Taiwan, Japan, and South Korea

Taiwan is similar to Japan and South Korea in terms of its reimbursement policy for medical devices by incorporating funding under technical fees and separate from technical fees. In addition, the pricing mechanism is set according to functional categories — namely, and existing functional category or a new functional category. In Taiwan, special devices classified under an existing functional category are given the lowest reimbursement point designated to an item in the same existing functional category, without additional premium mechanisms; unless they are classified into "improved functional category" and pricing are assigned based on the treatment course-expense ratio method or an existing special device in a similar function category. Several factors may be considered to determine the premium rate, including enhanced clinical efficacy, greater safety for users, improvements in treatment procedures, invasiveness reduction, cost savings, ease of use, and treatment for patients with a rare disease. Japan and South Korea also have the premium function, but they follow different premium criteria. For example, in Japan, once a similar function item exists, the similar function category comparison method is used. Several different premium rates exist: the epochal function premium, utility premium, improvement premium, and orphan premium. Meanwhile, in South Korea, if the evidence of clinical usefulness submitted by the manufacturer is based on clinical evidence or technological evidence, a premium is granted. If the medical device is classified under the innovative function-

al category, some different pricing mechanisms occur among three countries. In Taiwan, pricing methods refer to international prices (the United States, Japan, Australia, and South Korea), purchase prices by public hospitals, self-paid fees, and the proposed price by manufacturers, among other factors. In South Korea, one of the pricing mechanisms refers to prices in other countries, which is similar to Taiwan. However, in Japan, the cost accounting method is used. The pricing mechanisms for medical devices in Taiwan, Japan, and South Korea are summarized in Tab. 2 (see Attachment 1).

Although Taiwan, Japan, and South Korea all have HTA system, however there are some medical devices remain require HTA evaluation before reimbursement approval decisions in Taiwan and South Korea, but not in Japan. However, the three Asia countries adopt clinical evidence at different points in the pricing decision-making process. For example, in Taiwan, the clinical evidence determines the appropriate balance-billing ratio and patient payment price. In Japan, manufacturers can submit new clinical data to request the creation of a new functional category or obtains a new premium price after the reimbursement listing. In South Korea, clinical evidence is used for applications requesting premium prices.

## Conclusions and future challenges

The number of medical devices covered by Taiwan's NHI has gradually increased while manufacturers' price launches are constantly increasing. Therefore, reimbursement policy and pricing must change over time as well. Although Taiwan, South Korea, and Japan still provide reimbursements for medical devices based on functional categories, new reimbursement coverage and pricing rules have been implemented. For example, the requirement of a premium price on a medical device is based on clinical evidence in South Korea, whereas in Japan, manufacturers can submit new clinical data to request the creation of a new functional category or premium price after the reimbursement is listed. Currently, some advanced medical devices fall under the balance-billing devices category in Taiwan — sometimes without clear health benefits and big price differential problems. Therefore, referring to South Korea's evidence-based pricing, Taiwan revised its "Operation Directions for Processing as a balance-billing item of NHI's Medical Devices" on February 24, 2020. The upper limit of the special device for the insurer's balance-billing is determined according to the clinical evidence level so that it is more secure for patients, for whom health insurance benefits can reach up to 40%. More clinical benefit evidence yields more health insurance benefits from NHI, which can be used as a reference for patients to help make decisions. However, this new policy still faces a lot of challenges. The hope is that revisions can provide a reasonable price mechanism for balance-billing items, and these new mechanisms are expected to provide a transparent and reasonable decision-making process, although careful monitoring of these new mechanisms is warranted to enhance the reimbursement decision-making process.

## Notes

Reviewer comments see attachment 2.

## Acknowledgements

The authors thank the former CDE/HTA colleague Mr. Yu Li for his valuable assistance in the literature collection and review. The authors thank the National Health Insurance Administration (NHIA), Ministry of Health and Welfare (MOHW) for their financial support (no. 1070077726). No funding support was provided for the preparation of this manuscript.

## Conflict of interest

The authors declare that they have no competing interests.

## Attachments

1. attachment1_hta000134.pdf (168 KB)
   Table 2: The pricing mechanisms for medical devices in Taiwan, Japan and South Korea

2. attachment 2_hta000134.pdf (205 KB)
   Reviewer Comments

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
