## [Reviewer Comments · GMS Health Innovation and Technologies]

Comment		Author Response
Reviewer 1		
-	Comment: The article highlights the reimbursement coverage system in Taiwan, added with slight comparison with neighboring countries. The information is sufficient with the title and aim of the study as a short review/short communication. Additional elaboration can be added before the conclusion to briefly summarize the similarities/differences between all three countries, and highlight Taiwan pov as a punchline, as Taiwan is the main focus on this article.	Thank you for pointing this out. We have made the appropriate changes within the manuscript. We have summarized the similarities /differences between all three countries before the conclusion. Please refer to "2.4 Summary of medical device reimbursement and pricing mechanisms in Taiwan, Japan, and South Korea" section.
Reviewer 2		
18.) Please summarize the main findings of the study.	Comment: The paper describes the new regulatory and reimbursement process for medical devices in Taiwan. The paper goes on to summarise the regulatory processes in Japan and Korea (should this be referred to as South Korea?).	Thank you for pointing this out. We have made the appropriate changes within the manuscript. We have summarized the similarities /differences between all three countries before the conclusion. Please refer to "2.4 Summary of medical device reimbursement and pricing mechanisms in Taiwan, Japan, and South Korea" section; and we have amended Korea to South Korea.
19.) Please highlight the limitations and strengths.	Comment 1:The title of this paper is not "a comparison of medical device regulation in Taiwan, Japan and Korea",	Thank you for pointing this out.

	and yet it tries to be comparative, but is merely presenting a summary of each country's regulatory processes without an in-depth discussion of the pros and cons of each country's approach. Comment 2: The reference to the health systems of Australia, Canada and the US is irrelevant. Comment 3: The paper has an Introduction, then outlines the different processes, then has a Conclusion. Perhaps the paper needs an Introduction, an outline of the process in Taiwan, followed by a discussion section that refers to the similarities and differences of the other 2 countries. Comment 4: However, as the regulatory and reimbursement process in Taiwan appears to be a very complex system to describe, I think the paper would benefit from just describing that system in detail, rather than confusing the reader with details about Japan and Korea, which then doesn't seem to be discussed in any great detail. Comment 5: An initial discussion as to why a different	Responds 1 and 4: In light of the limited information about the reimbursement coverage and pricing rules of the medical device from Taiwan. Taiwan, Korea and Japan have similar reimbursement coverage decisions, and attempted to amend the pricing mechanism is based on clinical evidence. Additionally, South Korea and Japan serve as reference countries for Taiwan. Therefore, this paper is mainly describing the process for determining the reimbursement policy and pricing mechanisms for medical devices in Taiwan; in addition, medical device decision-making processes and pricing systems in Korea and Japan, which have similar reimbursement coverage decisions as Taiwan, will also be briefly described in this manuscript rather than comparative three Asia country. Responds 2: This manuscript reference to the health systems of Australia, Canada and the US to provide the overview of the different insurance system among the countries; and three Asia countries adopt universal
--	---	--

	approach to regulation and reimbursement is needed for medical devices compared to pharmaceuticals may be beneficial. This may be useful especially when the paper refers to billing ratios depending on the level of evidence - with RCTs attracting 40% from NHI and case series only 20%. It would be rare RCTs to be conducted for many (most) devices. Comment 6: There are several concepts that are not clearly explained to the reader and would benefit from clarification and definition 1) the use of the term "new function" and "new functional categories"2) an example may be useful in the discussion of existing and new functional categories eg bioresorbable vascular scaffold may be considered an innovative functional category, whereas a coronary stent with a new drug in it might be considered an improved functional category? Comment 7: 3) special materials, special devices and special material devices appear to be interchangeable. The term needs to be defined and their use needs to be consistent.	healthcare system to cover almost all medical expense. Responds 3: We have briefly summarized the similarities / differences between all three countries before the conclusion. Please refer to "2.4 Summary of medical device reimbursement and pricing mechanisms in Taiwan, Japan, and South Korea" section. Responds 5: Medical devices are being developed more rapidly than pharmaceuticals, yet evidence of real clinical efficacy is difficult to obtain in a short time; for example, the true efficacy of a bioresorbable vascular scaffold cannot be demonstrated until 3 years after implantation. Related benefits are also difficult to reflect in clinical evidence (e.g., safer for user, improvements in treatment procedures), and the classification is more complicated than pharmaceuticals. Therefore, it is inappropriate for medical devices and pharmaceuticals to be governed by the same rules. Please refer to the second
--	---	--

	Comment 8: The paper states that there are "10 special materials"....does this mean that are 10 special materials devices currently approved? The statement that follows that regarding items not covered by NHI should be made earlier, if at all Comment 9: 4) A definition of the 2 main payment systems for medical devices would be helpful - fee for service and DRG - 5) for an unfamiliar reader, it is unclear what "balanced-billing" means. Is that the insurance paying some of the cost, and the patient funding a portion? When discussing balanced billing, there are 5 requirements - it's not clear if only one of these requirements or all 5 must be fulfilled.	paragraph of "1. Introduction" section. Responds 6: We have further explained the definition of "new functional categories" and give an example. Please refer to the third paragraph of "2.1 Taiwan" section. Responds 7: We have amended special materials to special devices. Responds 8: We have update balance-billing items as follow "As of this writing, Taiwan distinguishes 9 categories of balance-billing items....". Please refer to the second paragraph of the "2.1 Taiwan" section. The sentence " Several items are not covered under the NHI program,..." was move forward, please refer to the second paragraph of the "2.1 Taiwan" section. Responds 9: We have further explained the definition of "fee for service", "DRG" and "balanced-billing". Please refer to the "annotation b and c of the Table 1" and second paragraph of "2.1 Taiwan"
--	--	---

		section. And for additional explanations "Balance-billing items must have evidence supporting that they meet at least one of the following criteria", please also refer to the second paragraph of "2.1 Taiwan" section.
21.) Please provide your detailed review report to the editor and authors.	Comment 1: The paper reads like the different sections have been written by different authors - there is a very stilted flow to it. The sections describing the processes in Japan and Korea are clearer and easier to understand. Although the cover letter has stated that an English speaker has edited the draft, I think the paper would benefit from being edited again - not just for correct English, but in order to make the narrative flow better. Comment 2: A detailed report is in Q18I really think that this paper would benefit from confining itself to just discussing the regulatory process in Taiwan in greater detail. Describing these complex systems is difficult and I think this paper has assumed a great deal of knowledge. A passing reference to the processes in Japan and Korea would be sufficient in the Discussion section.	Thank you for pointing this out. We have made the appropriate changes within the manuscript. Responds 1: We had the entire manuscript revised by a native English-speaking profession editor as the reviewer suggested. Responds 2 and 4 : The detailed response has been explained in the previous paragraph (Q19). Responds 3: We have further explained the Table 1 and Table 2. Please refer to the third paragraph of "1. Introduction" and "2.4 Summary of medical device reimbursement and pricing mechanisms in Taiwan, Japan, and South Korea" section. Reference countries in Table 1, we have moved to the fourth paragraph of "2.2 Japan" section and second paragraph of "2.4 Summary of medical device

	Comment 3: Although the tables list the attributes of each country's healthcare systems and pricing mechanisms, there is no discussion about what this information actually means and no referral to tables in the paper. Comment 4: To a reader unfamiliar with the health systems of these countries, this information is somewhat meaningless. I would recommend an almost complete rewrite - keeping in mind what the key message of the paper is i.e. describing the new regulatory process for devices in Taiwan. It may be worth writing a follow-up paper that actually compares the 3 country's systems, referring back to this paper.	reimbursement and pricing mechanisms in Taiwan, Japan, and South Korea" section.
--	---	--